# Effect of Dietary Supplements Which Upregulate Nitric Oxide on Walking and Quality of Life in Patients with Peripheral Artery Disease: A Meta-Analysis

**DOI:** 10.3390/biomedicines11071859

**Published:** 2023-06-29

**Authors:** Shannon A. Wong, Aaron Drovandi, Rhondda Jones, Jonathan Golledge

**Affiliations:** 1Queensland Research Centre for Peripheral Vascular Disease, College of Medicine and Dentistry, James Cook University, Townsville, QLD 4811, Australia; shannon.wong@my.jcu.edu.au (S.A.W.); a.drovandi@leeds.ac.uk (A.D.); rhondda.jones@jcu.edu.au (R.J.); 2The Department of Vascular and Endovascular Surgery, Townsville University Hospital, Townsville, QLD 4814, Australia

**Keywords:** peripheral arterial disease, dietary supplements, nitric oxide, physical functional performance, quality of life

## Abstract

This systematic review pooled evidence from randomised controlled trials (RCTs) on the effectiveness of dietary upregulators of nitric oxide (NO) in improving the walking and quality of life of patients with peripheral artery disease (PAD). RCTs examining the effect of dietary upregulators of NO in patients with PAD were included. The primary outcome was the maximum walking distance. Secondary outcomes were the initial claudication distance, the six-minute walking distance, quality of life, the ankle-brachial pressure index (ABI), adverse events and risk of mortality, revascularisation or amputation. Meta-analyses were performed using random effects models. The risk of bias was assessed using Cochrane’s ROB-2 tool. Leave-one-out and subgroup analyses were conducted to assess the effect of individual studies, the risk of bias and intervention type on pooled estimates. Thirty-four RCTs involving 3472 participants were included. Seven trials tested NO donors, nineteen tested antioxidants, three tested NO synthase inducers and five tested enhancers of NO availability. Overall, the dietary supplements significantly improved the initial claudication (SMD 0.34; 95%CI 0.04, 0.64; *p* = 0.03) but not maximum walking (SMD 0.13; 95%CI −0.17, 0.43; *p* = 0.39) distances. Antioxidant supplements significantly increased both the maximum walking (SMD 0.36; 95%CI 0.14, 0.59; *p* = 0.001) and initial claudication (SMD 0.58; 95%CI 0.26, 0.90; *p* < 0.001) distances. The dietary interventions did not improve the physical function domain of the Short Form-36 (SMD −0.16; 95%CI −0.32, 0.00; *p* = 0.38), ABI or risk of adverse events, mortality, revascularisation or amputation. Dietary NO upregulators, especially antioxidants, appear to improve the initial claudication distance in patients with PAD. Larger high-quality RCTs are needed to fully examine the benefits and risks of these treatments. PROSPERO Registration: CRD42022256653.

## 1. Introduction

The most well-recognised symptom of peripheral artery disease (PAD) is intermittent claudication, which leads to a reduced walking ability, an impaired health-related quality of life and reduced physical activity, contributing to functional decline and an increased risk of major adverse cardiovascular events [1,2]. While the 2019 European Society of Vascular Medicine guidelines recommend considering cilostazol and naftidrofuryl in patients with intermittent claudication with substantially limited quality of life and inability to participate in walking training (Class IB recommendation), these agents have limited efficacy and frequent side effects and are not available in many countries [3,4]. Novel treatments are thus needed.

PAD is associated with microvascular dysfunction resulting from impaired production and bioavailability of nitric oxide (NO) [5]. NO regulates the vascular smooth muscle function and blood vessel tone and has been proposed to promote revascularisation of ischaemic limbs through stimulating both arteriogenesis and angiogenesis [2]. In healthy people, NO is produced endogenously by endothelial nitric oxide synthase (NOS) in response to elevated shear stress [6]. Patients with PAD lack the ability to endogenously increase vascular NO production due to endothelial dysfunction and oxidative stress [6]. Thus, treatments which exogenously upregulate the production of NO may be an effective means to promote revascularisation in patients with PAD, thereby reducing exertional pain and improving walking.

There are four main methods by which a dietary intervention might upregulate NO (Figure 1). During ischemia, nitrate and nitrite can act as a reservoir of NO as they can be rapidly reduced to NO via the nitrate–nitrite–NO pathway (Figure 1) [7,8]. Increases in plasma nitrite can be achieved by dietary supplementation of inorganic nitrates which are reduced to nitrites by gut bacteria and converted to NO in the acidic environment of the small intestine [2]. Multiple studies have reported that nitrate supplementation decreases the oxygen cost of exercise [9,10] and improves physical function [9,10,11,12], with the greatest effects in populations with low aerobic capacity [13,14,15,16,17], including patients with PAD. NO production can also be increased via the citrulline–arginine–NO pathway (Figure 1). Supplementing L-arginine increases the substrates for endothelial NO synthase (NOS) [18]. Creatinine supplementation also increases NO availability along this pathway, as it reduces the activity of L-arginine:glycine amidinotransferase, thereby increasing the availability of L-arginine for NO synthesis [18]. NO production can also be increased by inducing the activity of endothelial NOS (Figure 1). Natural flavonol- and polyphenol-rich foods such as cocoa beans activate endothelial NOS and improve vascular function in patients with cardiovascular disease [19,20]. Finally, antioxidants can decrease the amount of superoxide present in vascular smooth muscles so that endogenous production of NO is enhanced (Figure 1) [21]. Common antioxidant supplements include flaxseed, garlic, Gingko biloba, polyunsaturated fatty acids, L-carnitine, N-acetylcysteine and vitamin E. Since individuals with PAD tend to have antioxidant deficiencies, it is plausible that antioxidant supplementation may improve NO production and PAD-related outcomes [22].

It is currently unclear whether dietary supplements aimed to upregulate NO are effective for treating PAD, with no consensus in practice guidelines [23,24,25]. The most recent American Heart Association PAD guidelines highlighted the need for studies to investigate the role of dietary interventions [25]. The objective of this systematic review and meta-analysis was to pool evidence from randomised controlled trials (RCTs) testing the efficacy of dietary supplementation aimed to upregulate NO in improving the walking distance and quality of life in patients with PAD.

## 2. Methods

### 2.1. Search Strategy and Eligibility Criteria

This systematic review was performed according to the Preferred Reporting Items for Systematic reviews and Meta-analyses (PRISMA) statement [26] and registered with PROSPERO (CRD42022256653). The completed PRISMA checklist can be found in Appendix A. Scopus (from 1966), MEDLINE (via OvidSP; from 1966) and CINAHL (from 1937) were searched from inception to 25 June 2022 to identify RCTs examining the effect of dietary upregulation of the NO pathway on walking distance and quality of life in patients with PAD. The search strategy included the key concepts of PAD, limb ischemia, intermittent claudication, nitrates, NO, antioxidants, dietary supplements and RCTs. The full search string is shown in Appendix A. No restrictions on language were applied. The reference lists of identified articles and reviews were also searched. Titles, abstracts and full texts were screened by two authors (S.A.W. and A.D.) to identify eligible studies, with disagreements resolved by consensus. For inclusion, studies needed to (a) be RCTs including participants with PAD diagnosed using recognised diagnostic criteria such as the ankle-brachial pressure index (ABI), (b) include an intervention group receiving dietary sources of NO upregulation, (c) include a control group not receiving dietary sources of NO pathway upregulation but otherwise receiving similar care and (d) assess at least one of the following outcomes: maximum walking distance, 6 min walk distance, intermittent claudication distance or quality of life measured by a validated questionnaire. Eligible dietary sources of NO pathway upregulation included four different categories: (a) NO donors, (b) enhancers of NO availability via the citrulline–arginine–NO pathway, (c) NOS inducers or (d) antioxidants (Figure 1). A dietary supplement was defined as an orally ingested product that contains a vitamin, mineral, herb, amino acid and/or other substances intended to supplement the diet. Studies were excluded if they tested a compound that was not administered orally, required a prescription to be accessed or required government approval to be made or sold.

### 2.2. Data Extraction

The primary outcome was maximum walking distance in a treadmill test. Secondary outcomes included initial claudication distance in either a treadmill test or corridor walking test, six-minute walking distance, quality of life assessed by validated tools including the Short Form (SF)-36 and Walking Impairment Questionnaire (WIQ), ABI and safety outcomes, including total adverse events, total serious adverse events, mortality and requirement for surgical or endovascular revascularisation or amputation. Adverse events were defined as any unintended and unfavourable sign or symptom temporally associated with the use of the intervention. Adverse events were considered serious if they resulted in mortality, threat to life, inpatient hospitalisation or prolongation of hospital stay or disruption of the ability to conduct normal activities of daily living. Other data extracted included age, sex, smoking, diabetes, inclusion and exclusion criteria, descriptions of intervention and control, assessments of adherence, follow-up duration and loss to follow-up. Loss to follow-up was defined as study participants for whom primary outcome data were not reported. Intention-to-treat results were used for the meta-analyses, where available. Missing data were to be resolved first by contacting the primary study investigators for unreported data, and then if unsuccessful, by imputation (Appendix A). Data were extracted by two independent reviewers, with disagreements resolved through discussion.

### 2.3. Quality Assessment

Risk of bias was assessed independently by two reviewers (S.A.W. and A.D.) using version 2 of the Cochrane risk of bias tool (ROB-2) [27]. Any disagreements were resolved by discussion until consensus was reached. Five domains were assessed to determine an overall risk of bias including randomisation process, deviation from intended interventions, missing outcome data, measurement of outcome and selection of reported results [27]. Risk of bias tables and plots [28] were created to summarise the risk of bias for the five individual domains and reported as “low risk”, “some risk” or “high risk” of bias. The assessment of these domains was used to judge the overall risk of bias for each included study. An RCT was deemed as “low risk” of bias if all five domains were judged to be at low risk, at “some risk” of bias if any of the domains were judged to be at some risk but none were judged to be at high risk and “high risk” of bias if any of the domains were judged to be at high risk [27].

### 2.4. Data Analysis

Data for all primary and secondary outcomes were synthesised in meta-analyses if data were reported for at least three trials. Meta-analyses were performed using the inverse variance method for continuous outcomes and Mantel–Haenszel’s statistical method for dichotomous outcomes with random effect models anticipating substantial heterogeneity. The results were reported as standardised mean differences (SMD) or odds ratios (OR) and 95% confidence intervals (CI). SMDs were interpreted as Cohen’s d values, with d = 0.2 regarded as a “small” effect size, d = 0.5 as a “medium” effect size and d = 0.8 as a “large” effect size. All statistical tests were two-sided and *p*-values < 0.05 were considered significant. Statistical heterogeneity was assessed using the I^2^ statistic and interpreted as low (0 to 49%), moderate (50 to 74%) or high (75 to 100%). Subgroup meta-analyses were conducted to assess the impact of risk of bias, specific types of dietary interventions and testing the primary outcome using fixed effects rather than a random effects model if there were at least two studies for any outcome measure. Leave-one-out (LOO) sensitivity analyses were performed to assess the contribution of each study to the effect estimates by excluding individual studies and recalculating the pooled estimates. Funnel plots comparing the summary estimate of each study and its precision (1/standard error) were used to assess publication bias [29]. All analyses were conducted using Review Manager 5 (RevMan 5) version 5.4 [30].

## 3. Results

### 3.1. Selection and Description of Included Studies

From 1611 unique records identified from the search strategy, 37 publications that reported on 35 trials were eligible for inclusion (Figure 2 and Appendix A) [31,32,33,34,35,36,37,38,39,40,41,42,43,44,45,46,47,48,49,50,51,52,53,54,55,56,57,58,59,60,61,62,63,64,65,66]. Of these, 28 trials were parallel and 7 were crossover trials. The outcomes of a trial assessing the efficacy of propionyl-L-carnitine were reported in two publications, including a report on the effect on walking distance [34] and another on quality of life [35]. The 35 eligible trials [31,32,33,34,35,36,37,38,39,40,41,42,43,44,45,46,47,48,49,50,51,52,53,54,55,56,57,58,59,60,61,62,63,64,65] included a total of 3538 randomised participants from eight countries across Europe, Asia, North America and South America, allocated to either dietary interventions (*n* = 1619) or control groups (*n* = 1853) (Table 1). Sample sizes ranged from 8 to 501 participants and follow-up durations ranged from one day to 24 months. All eligible trials recruited adult participants with documented PAD, and 28 trials required participants to also have intermittent claudication. The exclusion criteria varied for the different trials (Appendix A).

### 3.2. Description of Interventions

The dietary interventions tested were diverse and spanned the four different categories as described in Figure 1—NO donors in seven trials [31,45,47,55,58,62,65], antioxidants in nineteen trials [32,33,34,36,37,38,39,40,42,43,44,46,48,49,51,57,59,60,63], NOS inducers in four trials [50,53,61,66] and enhancers of NO availability in five trials (Appendix A) [41,52,54,56,64]. Twenty-six RCTs had control groups that received a matched placebo that was identical in all aspects and nine used a non-identical placebo but otherwise similar management to the intervention group. A walking assessment consisting of treadmill and/or corridor walking tests was the primary outcome for most trials and was assessed as a secondary outcome in three trials [49,55,59]. Quality of life was assessed as a primary outcome in two trials and as a secondary outcome in nine other trials, with most studies using the SF-36 assessment. Other outcomes included in the trials are outlined in Appendix A.

### 3.3. Risk of Bias

Overall, 10 trials were deemed as low risk of bias, 20 as some risk of bias and 5 as high risk of bias (Figure 3 and Appendix A). The domains in which the RCTs were most commonly judged to have a high risk of bias were deviations from intended interventions, the randomisation process and missing outcome data (Appendix A).

### 3.4. Effectiveness of Dietary Interventions

Each trial reported on a varying number of outcomes (Appendix A).

#### 3.4.1. Walking Distance

##### Maximum Walking Distance

A meta-analysis of 18 trials (intervention *n* = 1057; control *n* = 1020) found that the dietary interventions had a small positive effect on maximum walking distance, but CIs crossed 0 (SMD 0.13; 95% CI −0.17, 0.43; *p* = 0.39, Figure 4) with a high degree of heterogeneity (I^2^ = 88%). There was a statistically significant improvement in the maximum walking distance in the intervention group compared to control when using fixed effect (SMD 0.18; 95% CI 0.10, 0.27; *p* < 0.0001) instead of random effect models (Appendix A). The funnel plot was asymmetrical (Appendix A). LOO sensitivity analyses suggested that removal of the Wilson 2007 study [64] reduced the heterogeneity and increased the effect size substantially (Appendix A). Subgroup analyses showed that studies with “some” risk of bias demonstrated the greatest effect of the intervention (Figure 4). In addition, only antioxidant interventions significantly improved the maximum walking distance compared to control, but all the antioxidant studies were deemed to have either some or high risk of bias (Appendix A). Six [33,34,37,39,43,51] of the eight trials that tested antioxidant interventions and reported the maximum walking distance used either propionyl-L-carnitine or L-carnitine, with the other two testing vitamin E [36] and a mitochondrial antioxidant called Mito-Q [57]. The only antioxidant studies that reported statistically significant improvements in the maximum walking distance tested L-carnitine and its derivatives [33,34,37,43,51].

##### Initial Claudication Distance

A meta-analysis of 23 trials (intervention *n* = 1143; control *n* = 1108) found that the initial claudication distance was significantly improved by dietary interventions (SMD 0.34; 95% CI 0.04, 0.64; *p* = 0.03, Figure 5) with a high degree of heterogeneity (I^2^ = 90%). The funnel plot was asymmetrical (Appendix A). LOO sensitivity analyses suggested that removal of the Santo 2006 trial [60] reduced the effect size substantially, while removal of the Wilson 2007 trial [64] increased the effect size substantially (Appendix A). Subgroup analyses showed that studies with “some” risk of bias demonstrated the greatest effect of the intervention (Figure 5). Only antioxidant interventions significantly improved initial claudication distance compared to control, but 12 out of the 13 antioxidant studies that reported initial claudication distance were deemed to have either some or high risk of bias (Appendix A). Eight [32,33,34,37,39,46,51,60] of the thirteen antioxidant RCTs that reported the initial claudication distance tested either propionyl-L-carnitine or L-carnitine, with the other five using Gingko biloba [42], garlic powder [48], Mito-Q [57], polyunsaturated fatty acids [59] and alpha-lipoic acid [63]. The only antioxidant studies that reported statistically significant improvements in the initial claudication distance tested L-carnitine and its derivatives [32,37,39,46,60].

##### Six-Minute Walking Distance

A meta-analysis of six trials (intervention *n* = 108; control *n* = 77) showed that dietary interventions did not have any effect on the 6-min walking distance (SMD 0.03; 95% CI −0.26, 0.33; *p* = 0.98, Appendix A), with a low degree of heterogeneity (I^2^ = 0%). The funnel plot was symmetrical (Appendix A). LOO sensitivity analyses suggested that removal of any individual trial did not affect the significance of this finding (Appendix A). Subgroup analyses did not show any difference in effect size between studies deemed as low risk of bias and those with some risk (Appendix A).

#### 3.4.2. Quality of Life

##### SF-36: Physical Function Domain

A meta-analysis of six trials (intervention *n* = 391; control *n* = 369) found that the physical function domain in the SF-36 was slightly decreased in the dietary intervention group (SMD −0.16; 95% CI −0.32, 0.00; *p* = 0.05, Figure 6) with a low degree of heterogeneity (I^2^ = 6%), but this finding did not reach statistical significance. LOO sensitivity analyses suggested that removal of the Goldenberg 2012 trial [43] made the finding statistically significant (Appendix A). The funnel plot was asymmetrical (Appendix A). There were insufficient studies to undertake a subgroup analysis for risk of bias or specific types of interventions.

##### WIQ: Walking Distance and Walking Speed Domains

A meta-analysis of six trials using the WIQ walking distance domain (intervention *n* = 194; control *n* = 187) demonstrated no significant effect of the dietary interventions on the perceived walking distance (SMD 0.04; 95% CI −0.16, 0.24; *p* = 0.71, Appendix A) with low heterogeneity (I^2^ = 0%). The funnel plot was symmetrical (Appendix A). The six trials that reported the WIQ walking speed domain (intervention *n* = 180; control *n* = 137) demonstrated no significant impact of the dietary interventions on the perceived walking speed (SMD 0.00; 95% CI −0.24, 0.23; *p* = 1.00, Appendix A) with low heterogeneity (I^2^ = 4%). The funnel plot was asymmetrical (Appendix A). In both walking distance and walking speed domains, LOO sensitivity analyses suggested that the removal of any individual trial did not affect the significance of the findings (Appendix A). In addition, subgroup analyses focused on the risk of bias did not change the significance of the outcome (Appendix A), and there were insufficient studies to undertake a subgroup analysis for intervention type.

#### 3.4.3. ABI

A meta-analysis of 11 trials (intervention *n* = 537; control *n* = 537) found that dietary interventions moderately improved the participants’ ABI when compared to control, though this increase was not statistically significant (SMD 0.31; 95% CI −0.19, 0.81; *p* = 0.23; I^2^ = 92%, Appendix A). The funnel plot was asymmetrical (Appendix A). LOO sensitivity analyses suggested that removal of any individual RCT did not affect the significance of the findings (Appendix A). Subgroup analyses demonstrated that the risk of bias and intervention type did not substantially affect the findings (Appendix A).

#### 3.4.4. Safety Outcomes

##### Adverse Events

A meta-analysis of 15 trials (intervention *n* = 1148; control *n* = 1127) found that dietary interventions increased the participants’ risk of adverse events, though this increase was not statistically significant (OR 1.32; 95% 0.76, 2.28; *p* = 0.33, Appendix A) with high heterogeneity (I^2^ = 78%). The funnel plot was asymmetrical (Appendix A). LOO sensitivity analyses suggested that removal of the Brevetti 1999 trial [33] made the finding statistically significant (Appendix A). The risk of adverse events was not influenced by the risk of bias or classification of intervention (Appendix A).

##### Serious Adverse Events

A meta-analysis of 16 trials (intervention *n* = 1126; control *n* = 1100) demonstrated that dietary interventions slightly decreased the participants’ risk of serious adverse events (OR 0.83; 95% CI 0.60, 1.16; *p* = 0.27, Appendix A) with low heterogeneity (I^2^ = 0%), but this finding was not statistically significant. The funnel plot was asymmetrical (Appendix A). LOO sensitivity analyses suggested that removal of any individual RCT had no effect on the significance of the findings (Appendix A). Subgroup analyses showed that studies with “some” risk of bias demonstrated the greatest effect of the intervention (Appendix A), but the intervention type had no effect on the risk of serious adverse events (Appendix A).

##### Mortality

A meta-analysis of 13 trials (intervention *n* = 874; control *n* = 882) found that dietary interventions had little effect on mortality (OR 1.14; 95% CI 0.49, 2.64; *p* = 0.76, Appendix A) with low heterogeneity (I^2^ = 0). The funnel plot was symmetrical (Appendix A). LOO sensitivity analyses suggested that removal of any individual trial did not affect the significance of the findings (Appendix A). There was an insufficient number of studies to conduct subgroup analyses to assess the impact of risk of bias or type of intervention on this outcome.

##### Requirement for Lower Extremity Revascularisation or Amputation

A meta-analysis of six trials (intervention *n* = 340; control *n* = 349) showed that dietary interventions had no significant effect on requirement of lower extremity revascularisation or amputation (OR 1.70; 95% CI 0.42–6.87; Z = 0.74, *p* = 0.46, Appendix A) with low heterogeneity (I^2^ = 0%). The funnel plot was asymmetrical (Appendix A). LOO sensitivity analyses found that removal of any individual trial did not affect the significance of the findings (Appendix A). There were insufficient studies to conduct subgroup analyses to assess the impact of risk of bias or type of intervention.

## 4. Discussion

This meta-analysis demonstrated that dietary interventions that upregulate the NO pathway moderately improved the initial claudication distance in patients with PAD, but the improvements in the maximum walking distance did not achieve statistical significance. The interventions classified as antioxidants had the greatest effect on walking distance. Quality of life, ABI and safety outcomes did not differ significantly between intervention and control groups.

Both the initial claudication and maximum walking distance on treadmill testing have been proposed as accurate measures of walking impairment in patients with PAD, although this remains controversial [67,68]. Few patients will actually walk until they reach their maximum pain threshold while performing normal daily activities [67]. The potential of dietary supplements to increase the initial claudication distance identified in this meta-analysis is encouraging for patients with intermittent claudication who have limited pharmacological options to improve their walking performance. The confidence in the benefit of dietary interventions is limited due to the marked statistical heterogeneity between studies, the lack of concordance in findings for the different walking distance outcomes, the lack of consistency in the findings of sensitivity and sub-group analyses and the frequent concerns of risk of bias identified within the included trials.

Antioxidants are known to protect against oxygen-free radicals and other toxic metabolites produced in ischemia and tissue damage [39] to enhance the production of endogenous NO. The effects of antioxidants on NO upregulation pathways in the endothelium and vascular smooth muscles may not be the only explanation for their superiority over other dietary interventions in improving walking performance. The only antioxidant trials in this meta-analysis that reported statistically significant improvements in walking distances studied L-carnitine and its derivatives [32,33,34,37,39,43,46,51,60]. L-carnitine and its derivatives have an important role in lipid metabolism, including the transport of long-chain fatty acids across the mitochondrial membrane to enable oxidation of fatty acids and conserve the coenzyme A pool to improve tissue energy metabolism [39]. Two studies assessed surrogate measures of carnitine, such as a muscle carnitine assay [32] and plasma carnitine concentrations [46], both of which demonstrated a significant increase in carnitine levels in the intervention group compared to the control. Further studies of L-carnitine and its derivatives are warranted, especially since the question remains if these compounds have a beneficial effect on walking distance through their effect on the NO pathway, given that the other dietary supplements that target the NO pathway are not as effective. None of the antioxidant studies measured NO levels. Further trials may consider assessing nitrate levels to determine the effect that antioxidants have on the NO pathway.

Interestingly, the studies with “some risk” of bias demonstrated greater effect estimates than those deemed “low risk” or “high risk”. Out of these studies, the ones that reported a significant effect of the intervention on maximum walking distance or initial claudication distance were mostly judged to have “some risk” of bias due to potential bias in the randomisation process or selection of reported results. These judgments were only due to a lack of information reported about the randomisation process or a pre-specified analysis plan, and not due to baseline differences between groups or issues with blinding or allocation concealment. Thus, the results of these studies demonstrating a positive effect of dietary nitrate interventions on the walking performance should not be dismissed. In addition, random-effects modelling reduces the effect of study size on the weightings of each study. A sub-analysis found that the primary outcome of maximum walking distance was significantly improved when using a fixed effect model, highlighting the need for larger high-quality trials.

The strengths of this systematic review include a pre-specified analysis plan, an extensive risk of bias assessment and reporting of sensitivity and subgroup analyses to evaluate the effects of individual studies, intervention types and risk of bias on pooled outcomes. There were several limitations to this meta-analysis. Firstly, there was significant heterogeneity amongst the included studies in terms of the interventions used and the study designs, populations and outcome assessments, which was reflected in the marked statistical heterogeneity for most outcomes. Although subgroup analyses were used to elucidate the most effective intervention types, these also had marked statistical heterogeneity. Most funnel plots suggested a risk of publication bias, and thus possible over-estimation of the interventions’ effects on the reported outcomes. Some of the LOO sensitivity analyses demonstrated the presence of outliers that could have skewed the results. Finally, meta-analyses of quality of life, ABI and safety outcomes were limited due to under-reporting.

## 5. Conclusions

This meta-analysis suggests that dietary interventions that upregulate NO may improve the initial claudication distance in patients with PAD. Antioxidant supplements, specifically L-carnitine and its derivatives, appeared to be the most effective interventions, and thus warrant further investigation. It remains unclear whether dietary supplements to upregulate NO influence patients’ quality of life, ABI or safety outcomes. Due to heterogeneity, risk of bias and inconsistent findings, further large and methodologically rigorous RCTs are needed to resolve the benefit of dietary NO upregulation and particularly antioxidants.

## Figures and Tables

**Figure 1 biomedicines-11-01859-f001:**
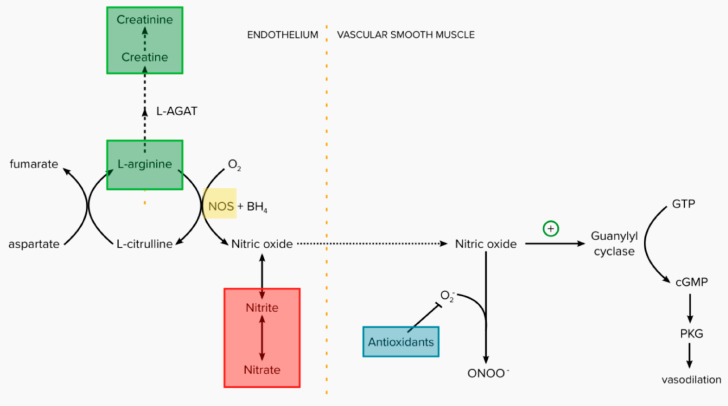
Potential targets for dietary supplements to increase nitric oxide production in patients with PAD. Targets include NO donors that increase substrates in the nitrate–nitrite–NO pathway (red), enhancers of NO availability which act on the citrulline–arginine–NO pathway (green), endothelial NOS inducers (yellow) and antioxidants (blue).

**Figure 2 biomedicines-11-01859-f002:**
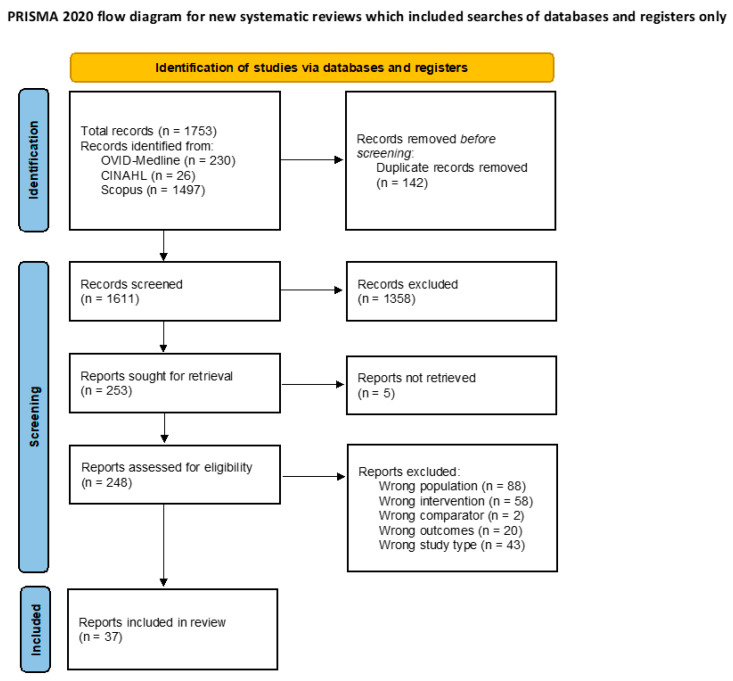
PRISMA flow diagram summarising the selection of eligible articles [26]. There were five full-text articles that were unable to be retrieved.

**Figure 3 biomedicines-11-01859-f003:**
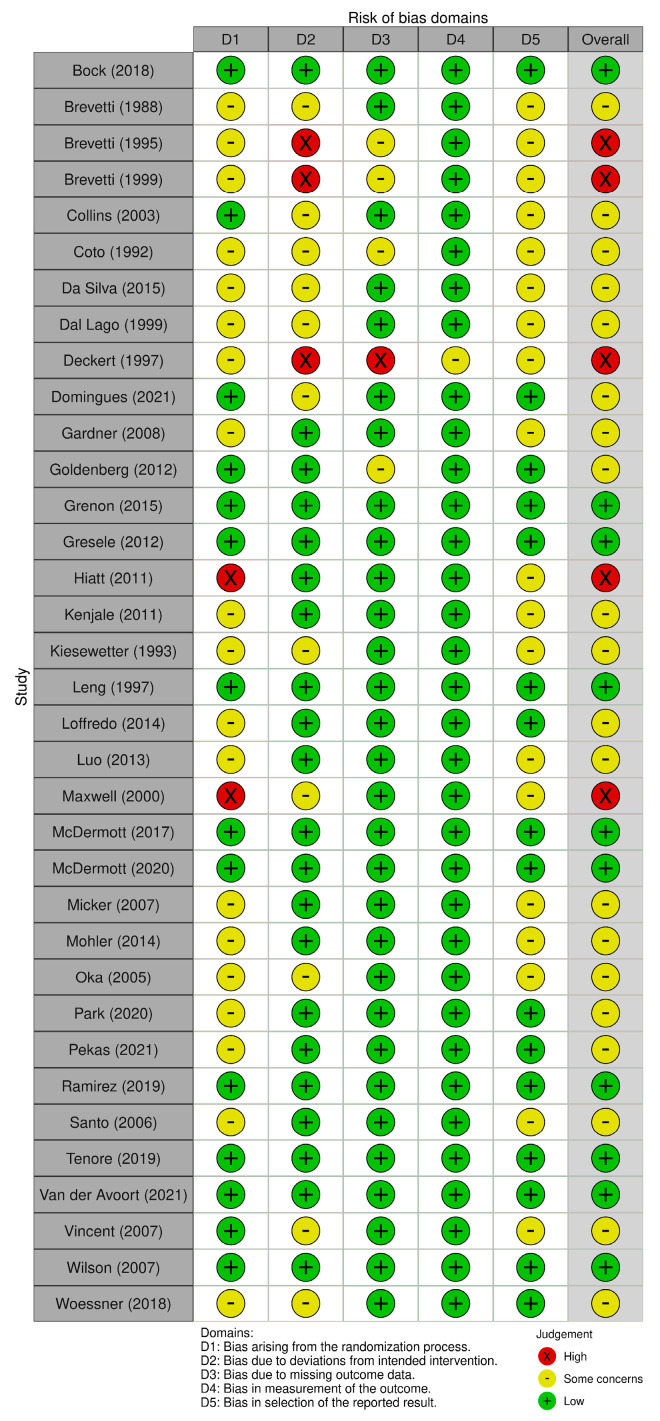
Risk of bias assessment of included trials [31,32,33,34,35,36,37,38,39,40,41,42,43,44,45,46,47,48,49,50,51,52,53,54,55,56,57,58,59,60,61,62,63,64,65,66] based on ROB2 tool [27].

**Figure 4 biomedicines-11-01859-f004:**
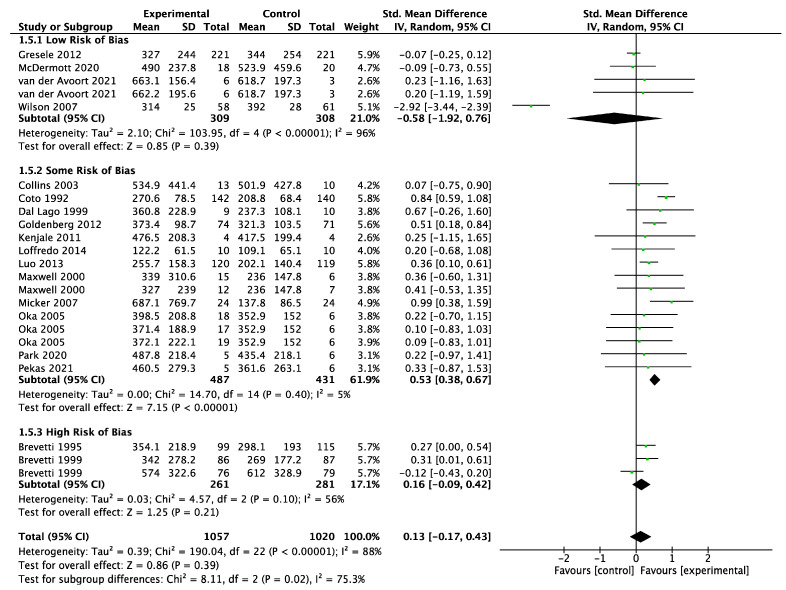
Forest plot showing the effect of dietary supplements upregulating the NO pathway versus control on the maximum walking distance; studies stratified by risk of bias [33,34,36,37,39,43,45,47,50,51,52,53,54,56,57,58,62,64].

**Figure 5 biomedicines-11-01859-f005:**
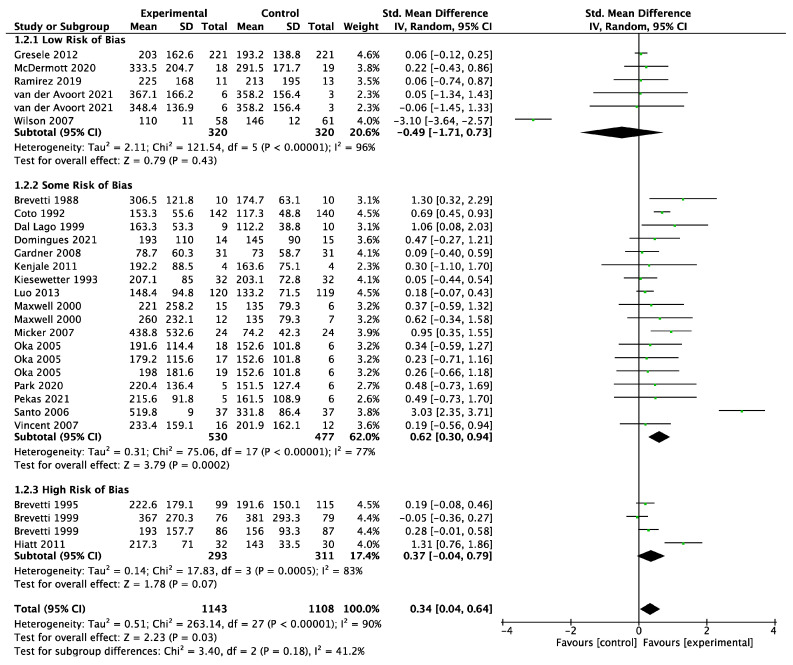
Forest plot showing the effect of dietary supplements upregulating the NO pathway versus control on the initial claudication distance; studies stratified by risk of bias [32,33,34,37,39,41,42,45,46,47,48,50,51,51,52,52,53,53,54,54,55,56,62,63,64].

**Figure 6 biomedicines-11-01859-f006:**
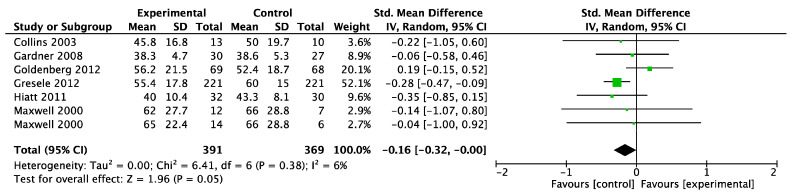
Forest plot showing the effect of dietary supplements upregulating the NO pathway versus control on quality of life measured by the physical function domain of the SF-36 assessment [36,42,43,45,46,52].

**Table 1 biomedicines-11-01859-t001:** Characteristics of included studies and participants at baseline.

								Intervention Group					Control Group				
Study	Country	Design	Total Participants	Attrition	Follow-Up Duration	Type of Intervention	Brief Population Description	N	Age	Female	Smoking	Diabetes	N	Age	Female	Smoking	Diabetes
Nitric Oxide Donors																	
Bock (2018) [31]	USA	DB RCT	21	0	2 months	1 g daily of sodium nitrate tablets	Adults with PAD aged 50–85 years with IC	13	73 (9)	7 (53.8%)	0 (0.0%)	4 (30.8%)	8	69 (10)	2 (25.0%)	0 (0.0%)	2 (25.0%)
Gresele (2012) [45]	Multiple European	DB RCT	442	70 (15.8%)	6 months	1.6 g daily of NCX 4016	Adults aged 40–80 years with PAD and IC	221	66.7 (8.2)	44 (19.9%)	91 (41.2%)	77 (34.8%)	221	66.4 (9.1)	48 (21.7%)	91 (41.2%)	67 (30.3%)
Kenjale (2011) [47]	USA	CO RCT	8	0	1 day	500 mL of nitrate rich beetroot juice	Adults with PAD and IC	8	67 (13)	4 (50.0%)	NR	NR	NA	NA	NA	NA	NA
Mohler (2014) [55]	USA	DB RCT	36	6 (16.7%)	3 months	80 mg or 160 mg daily of sodium nitrite tablets	Adults aged 35–85 years with PAD	19	65.3 (8.9)	4 (21.1%)	7 (36.8%)	12 (63.2%)	18	64.9 (9.0)	5 (27.8%)	7 (38.9%)	10 (55.6%)
								18	67.9 (10.0)	5 (27.8%)	2 (11.1%)	12 (66.7%)					
Pekas (2021) [58]	USA	CO DB RCT	11	0	1 day	Single dose of body-mass normalized beetroot juice	Adults with PAD and IC	11	70.0 (7.0)	6 (54.5%)	1 (9.1%)	3 (27.3%)	NA	NA	NA	NA	NA
Van der Avoort (2021) [62]	Netherlands	CO RCT	18	0	1 month	150 g nitrate rich vegetables, 70 mL nitrate-rich beetroot juice	Adults with PAD and stable IC for >3 months	18	73 (8)	7 (38.9%)	5 (27.8%)	4 (22.2%)	NA	NA	NA	NA	NA
Woessner (2018) [65]	USA	DB RCT	35	11 (31.4%)	3 months	Beetroot juice and a 36-session exercise program	Adults with PAD aged 40–80 years and stable IC pain	11	67.5 (8.6)	2 (18.2%)	4 (36.4%)	6 (54.5%)	13	71.5 (7.3)	7 (53.8%)	5 (38.5%)	2 (15.4%)
Enhancers of NO Availability																	
Domingues (2021) [41]	Brazil	DB RCT	32	3 (9.4%)	2 months	5 g daily creatinine monohydrate (after 20 g/day for 7 days)	Adults with PAD and IC	14	64 (10)	6 (42.9%)	11 (78.6%)	7 (50.0%)	15	64 (8)	8 (53.3%)	12 (78.6%)	9 (60.0%)
Maxwell (2000) [52]	USA	DB RCT	41	1 (2.4%)	2 weeks	One or two 50 g L-arginine-enriched nutrient bars daily	Adults with PAD and stable IC	15	68.2 (7.7)	1 (6.7%)	2 (13.3%)	2 (13.3%)	14	70.6 (7.5)	5 (35.7%)	4 (28.6%)	4 (28.6%)
								12	66.3 (10.4)	4 (25.0%)	0	2 (16.7%)					
Micker (2007) [54]	Poland	DB RCT	48	0	1 month	12 g daily L-arginine tablets	Adults with PAD	24	59.1 (8.8)	12 (50.0%)	11 (45.8%)	0	24	66.0 (7.6)	11 (45.8%)	13 (54.2%)	0
Oka (2005) [56]	USA	DB RCT	80	8 (10.0%)	3 months	3 g, 6 g, or 9 g daily of L-arginine tablets	Adults aged at least 40 years with PAD and IC	18	75 (9)	6 (33.0%)	2 (11.1%)	11 (61.1%)	18	72 (9)	3 (16.7%)	2 (11.1%)	1 (5.6%)
								17	76 (6)	6 (35.3%)	2 (11.8%)	6 (35.3%)					
								19	73 (6)	7 (36.8%)	1 (5.3%)	7 (36.8%)					
Wilson (2007) [64]	USA	DB RCT	133	14 (10.5%)	6 months	1 g three times daily of L-arginine tablets	Adults with PAD aged at least 45 years with stable IC pain	66	73 (9)	14 (21.2%)	8 (12.1%)	20 (30.3%)	67	72 (7)	18 (26.9%)	12 (17.9%)	20 (29.9%)
Nitric Oxide Synthase Inducers																	
Loffredo (2014) [50]	USA	CO SB RCT	20	0	1 day	Single dose of 40 g of dark chocolate	Adults with PAD and IC	20	69 (9)	6 (30.0%)	0	6 (30%)	NA	NA	NA	NA	NA
McDermott (2017) [66]	USA	DB RCT	66	2 (3.0%)	6 months	125 mg or 500 mg of resveratrol daily	Adults aged at least 65 years with PAD	44	74.6 (7.0)	14 (31.8%)	9 (20.5%)	12 (27.3%)	22	74.1 (6.1)	7 (31.8%)	5 (22.7%)	14 (63.6%)
McDermott (2020) [53]	USA	DB RCT	44	4 (9.1%)	6 months	Three flavanol-rich packets mixed with water daily	Adults aged at least 60 years with PAD	23	71 (7)	8 (34.8%)	11 (47.8%)	13 (56.5%)	21	73 (7)	7 (33.3%)	3 (14.3%)	11 (52.4%)
Tenore (2019) [61]	Italy	DB RCT	180	0	6 months	1 g twice daily of Annurca apple polyphenolic extract	Adults with PAD aged 35 to 75 years with IC pain	90	71.5 (9.4)	28 (31.1%)	0	20 (22.2%)	90	70.5 (10.2)	22 (24.4%)	0	24 (26.7%)
Antioxidants																	
Brevetti (1988) [32]	Italy	CO DB RCT	20	0	3 weeks	4 g daily of L-carnitine	Adults with PAD and IC	20	59.8 (7.0)	0	NR	NR	NA	NA	NA	NA	NA
Brevetti (1995) [34]	Italy	DB RCT	245	31 (12.7%)	6 months	3 g daily (via up-titration) of propionyl-L-carnitine	Adults with PAD aged at least 40 years	118	61.8 (7.6)	8 (6.8%)	114 (96.6%)	17 (14.4%)	127	58.9 (7.9)	13 (10.2%)	120 (94.5%)	26 (20.5%)
Brevetti (1999) [33]	Multiple European	DB RCT	501	173 (34.5%)	12 months	2 g daily of propionyl-L-carnitine	Adults with PAD	239	61.9 (8.8)	38 (15.9%)	208 (87.0%)	12 (5.0%)	246	62.7 (8.9)	51 (20.7%)	209 (85.0%)	12 (4.9%)
Collins (2003) [36]	USA	DB RCT	25	2 (8.0%)	6 months	400IU daily of Vitamin E	Adults with PAD and IC	13	67.2 (9.4)	0	6 (46.2%)	NR	12	70.2 (8.3)	0	3 (25.0%)	NR
Coto (1992) [37]	Italy	DB RCT	300	18 (6.0%)	6 months	2 g daily of propionyl-L-carnitine capsules	Adults with PAD and IC limiting walking ability	150	60.6 (6.9)	62 (41.3%)	NR	41 (27.3%)	150	60.3 (6.5)	55 (36.7%)	34 (22.7%)	NR
Da Silva (2015) [38]	Brazil	CO DB RCT	11	1 (9.1%)	4 days	1.8 g daily of N-acetylcysteine effervescent tablets	Adults with PAD and IC	10	62 (6.3)	0	6 (60.0%)	4 (40.0%)	NA	NA	NA	NA	NA
Dal Lago (1999) [39]	Italy	DB RCT	22	2 (9.1%)	3 months	3 g daily of propionyl-L-carnitine	Adults with PAD aged 40–75 years with IC	11	NR	NR	NR	0	11	NR	NR	NR	0
Deckert (1997) [40]	Italy	DB RCT	245	58 (23.7%)	6 months	1 g to 3 g daily or propionyl-L-carnitine	Adults aged at least 40 years with PAD and IC	NR	NR	NR	NR	NR	NR	NR	NR	NR	NR
Gardner (2008) [42]	USA	DB RCT	62	6 (9.7%)	4 months	300 mg daily of EGb 761 (Ginkgo Biloba)	Adults with PAD and IC	31	70 (8)	7 (22.6%)	6 (19.4%)	13 (41.9%)	31	69 (8)	18 (58.1%)	4 (12.9%)	18 (58.1%)
Goldenberg (2012) [43]	USA	DB RCT	163	34 (20.9%)	6 months	2 g daily L-carnitine plus 200 mg daily cilostazol	Adults aged at least 40 years with PAD and IC	74	65.8 (9.4)	12 (16.2%)	32 (43.2%)	NR	71	67.3 (8.0)	15 (21.1%)	23 (32.4%)	NR
Grenon (2015) [44]	USA	DB RCT	80	8 (10.0%)	1 month	4.4 g daily of n3-PUFA (fish oil)	Adults aged at least 50 years with PAD and IC	40	68 (7)	1 (2.5%)	38 (95.0%)	11 (27.5%)	40	69 (9)	1 (2.5%)	36 (90.0%)	14 (35.0%)
Hiatt (2011) [46]	USA	DB RCT	69	10 (14.5%)	6 months	2 g daily of propionyl-L-carnitine and exercise therapy	Adults aged 40–80 years with PAD and IC	32	67.4 (8.7)	12 (37.5%)	12 (37.5%)	3 (9.4%)	30	66.6 (8.8)	5 (16.7%)	5 (16.7%)	10 (33.3%)
Kiesewetter (1993) [48]	Germany	DB RCT	80	16 (20.0%)	3 months	800 mg daily of garlic powder	Adults aged 40–75 years with PAD and stable IC	32	59.9 (10.6)	9 (28.1%)	24 (75%)	5 (15.6%)	32	60.1 (7.7)	12 (37.5%)	24 (75%)	8 (25%)
Leng (1997) [49]	UK	DB RCT	120	45 (37.5%)	24 months	Multi-ingredient antioxidant capsules	Adults with PAD and stable IC	55	66.2 (7.0)	19 (34.5%)	21 (38.2%)	5 (9.1%)	65	65.3 (7.3)	19 (29.2%)	26 (40.0%)	6 (9.2%)
Luo (2013) [51]	China	DB RCT	239	23 (9.6%)	4 months	2 g daily of propionyl-L-carnitine	Adults aged 40–75 years with PAD and stable IC	120	NR	NR	NR	NR	119	NR	NR	NR	NR
Park (2020) [57]	USA	CO RCT	11	0	1 day	Single dose of 80 mg MitoQ	Adults with PAD and IC	11	66.1 (10.6)	6 (54.5%)	NR	2 (18.9%)	NA	NA	NA	NA	NA
Ramirez (2019) [59]	USA	DB RCT	24	4 (16.7%)	3 months	2.2 g twice daily of n-3 polyunsaturated fatty acid	Adults with PAD aged at least 50 years with IC pain	11	69 (8)	0	10 (90.9%)	3 (27.3%)	13	73 (7)	0	13 (100%)	4 (30.8%)
Santo (2006) [60]	Italy	DB RCT	74	0	12 months	2 g daily of propionyl L-carnitine	Adults with PAD and type 2 diabetes	37	61.8 (3.0)	NR	0	37 (100%)	37	61.3 (1.6)	NR	0	37 (100%)
Vincent (2007) [63]	USA	DB RCT	32	4 (12.5%)	3 months	300 mg twice daily of alpha-lipoic acid capsules	Adults with PAD aged at least 50 years with IC pain	16	75.1 (8.2)	7 (43.8%)	0	4 (25.0%)	12	70.7 (18.9)	6 (50.0%)	0	3 (25.0%)

Data are presented as *n* (%), mean (standard deviation), or median [range or IQR]. DB = double-blind; CO = cross-over; IC = intermittent claudication; NA = not applicable (cross-over trial); NR = not reported; PAD = peripheral artery disease; RCT = randomized controlled trial; SB = single blind.

## Data Availability

All the data presented are contained within the article or Appendix A.

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
