# Peer review of "Effect of Dietary Supplements Which Upregulate Nitric Oxide on Walking and Quality of Life in Patients with Peripheral Artery Disease: A Meta-Analysis"

_biomedicines, 2023, doi:10.3390/biomedicines11071859_

Round 1
Reviewer 1 Report
Wong and colleagues present a systematic review on dietary nitrate supplements in patients with peripheral artery disease.
I have only some minor comments, which should be adressed before publication:
1. Please use instead ABPI the more common term ABI.
2. Amongst others, cilostazol, naftidrofuryl, pentoxifylline, buflomedil, carnitine and propionyl-L-carnitine were studied for improvement of walking distance, see also ESC guidelines for PAD (2018)
The use of cilostazol or naftidrofuryl both have a Ib recommendation in the ESVM guidelines on peripheral artery disease (2019). This should be mentioned and cited.
3. Please be aware that citations are included at the end of the sentences (some citations are missing).
4. Please include the p- values in the abstract.
5. Could you comment on Fontaine/ Rutherford classification of the patients?
7. The quality of the figures and tables should be improved.
Author Response
- Please use instead ABPI the more common term ABI.
- I have replaced “ABPI” with “ABI” throughout the manuscript.
- Amongst others, cilostazol, naftidrofuryl, pentoxifylline, buflomedil, carnitine and propionyl-L-carnitine were studied for improvement of walking distance, see also ESC guidelines for PAD (2018). The use of cilostazol or naftidrofuryl both have a Ib recommendation in the ESVM guidelines on peripheral artery disease (2019). This should be mentioned and cited.
- I have added this line in the introduction and cited the ESVM guidelines as suggested – “While the 2019 European Society of Vascular Medicine guidelines recommend considering cilostazol and naftidrofuryl in patients with intermittent claudication with substantially limited quality of life and inability to participate in walking training (Class IB recommendation), these agents have limited efficacy and frequent side effects and are not available in many countries.”
- Please be aware that citations are included at the end of the sentences (some citations are missing).
- I have now included additional citations throughout the manuscript.
- Please include the p- values in the abstract.
- I have now included p-values in the abstract as suggested.
- Could you comment on Fontaine/ Rutherford classification of the patients?
- This data was not consistently reported in the papers and was thus not collected for my analysis.
- The quality of the figures and tables should be improved.
- I will upload higher resolution figures/tables for publication.
Reviewer 2 Report
Effect of dietary supplements which upregulate nitric oxide on walking and quality of life in patients with peripheral artery disease: a meta-analysis
It is a meta-analysis of studies with food supplements regulating NO and potential effect on PAD in humans. It is good that these meta-analyses are performed, taking bias in consideration, otherwise one keeps producing data and initiate novel clinical studies without focus. In my view it becomes clear now that L-carnitine and its derivatives deserve most attention for further investigation. Moreover, the question remains if the latter compounds have their beneficial effect through the improved NO pathway, since other NO targeting food supplements are not as effective. Is that also how the authors see it? It would be good if the authors can take a more dedicated stand on what they think should be done next, and not keep everything open.
Another example of this is the food supplement resveratrol, with multiple functions including improving NO function (Nitric Oxide. 2022 Dec 1;129:74-81. doi: 10.1016/j.niox.2022.10.005.) is also not beneficial in a human study on PAD. Is the RESTORE trial helpful to either discuss or could it be added to the meta-analysis if it meets the criteria. (Restore trial: JAMA Cardiol. 2017 Aug 1;2(8):902-907. doi: 10.1001/jamacardio.2017.0538.)
I especially like it that the authors stress the importance of evaluation of study bias.
Perhaps the results can be portrayed in a graphic way as a visually attractive figure?
Author Response
It is a meta-analysis of studies with food supplements regulating NO and potential effect on PAD in humans. It is good that these meta-analyses are performed, taking bias in consideration, otherwise one keeps producing data and initiate novel clinical studies without focus. In my view it becomes clear now that L-carnitine and its derivatives deserve most attention for further investigation. Moreover, the question remains if the latter compounds have their beneficial effect through the improved NO pathway, since other NO targeting food supplements are not as effective. Is that also how the authors see it? It would be good if the authors can take a more dedicated stand on what they think should be done next, and not keep everything open.
Response:
- I have added this line in the discussion: “Further studies of L-carnitine and its derivatives are warranted, especially since the question remains if these compounds have a beneficial effect on walking distance through their effect on the NO pathway, given that the other dietary supplements that target the NO pathway are not as effective.”
- I have also written this in the conclusion: “Antioxidant supplements, specifically L-carnitine and its derivatives, appeared to be the most effective interventions, and thus warrant further investigation.”
Another example of this is the food supplement resveratrol, with multiple functions including improving NO function (Nitric Oxide. 2022 Dec 1;129:74-81. doi: 10.1016/j.niox.2022.10.005.) is also not beneficial in a human study on PAD. Is the RESTORE trial helpful to either discuss or could it be added to the meta-analysis if it meets the criteria. (Restore trial: JAMA Cardiol. 2017 Aug 1;2(8):902-907. doi: 10.1001/jamacardio.2017.0538.)
Response:
- Thank you for mentioning this trial – I think this trial fits criteria so I will work on adding it into the meta-analysis
I especially like it that the authors stress the importance of evaluation of study bias. Perhaps the results can be portrayed in a graphic way as a visually attractive figure?
Response:
- We have already included (1) risk of bias of each study in the traffic light plot, and (2) the outcomes stratified by study bias in figures/supplementary figures
- Are there other specific results that the reviewer would want to see in a figure?
Reviewer 3 Report
The meta analysis titled "Effect of dietary supplements which upregulate nitric oxide on walking and quality of life in patients with peripheral artery disease: a meta-analysis" is a well written paper, in general.
Methods for writing a systematic review is followed by the authors. Methodology and presentation of the results are comprehensive and fine.
Discussion is fair as the conclusions.
However, all of the figures are too blurry and they should be replaced with figures with higher resolution. Otherwise, scientific quality of the meta-analysis is high and can be published after revision.
Author Response
The meta analysis titled "Effect of dietary supplements which upregulate nitric oxide on walking and quality of life in patients with peripheral artery disease: a meta-analysis" is a well written paper, in general. Methods for writing a systematic review is followed by the authors. Methodology and presentation of the results are comprehensive and fine.
Discussion is fair as the conclusions. However, all of the figures are too blurry and they should be replaced with figures with higher resolution. Otherwise, scientific quality of the meta-analysis is high and can be published after revision.
Response:
Thank you for the feedback – I will be sure to include higher resolution figures for publication.